# Association between self-reported napping and risk of cardiovascular disease and all-cause mortality: A meta-analysis of cohort studies

**Meng Wang[1], Xin Xiang[2], Zhengyan Zhao[3], Yu liu[4], Yang Cao[1], Weiwei Guo[1], Linlin Hou[5], Qiuhuan Jiang[5]***

1 Department of Nursing, Faculty of Medicine & Health Sciences, Universiti Putra Malaysia, Serdang, Selangor, Malaysia, 2 College of Acupuncture and Massage, Changchun University of Chinese Medicine, Changchun City, Jilin Province, China, 3 Department of Endocrinology, Zhengzhou Seventh People's Hospital, Zhengzhou City, Henan Province, China, 4 Emergency Medicine Department of the Second Mobile Contingent Hospital of the Chinese People's Armed Police Forces, Wuxi City, Jiangsu Province, China, 5 Henan Provincial People's Hospital, Zhengzhou City, Henan Province, China

* qiuhuan1890@163.com

**Data Availability Statement:** All relevant data are within the manuscript and its Supporting information files.

## Abstract

### Objectives

This meta-analysis aims to assess the association between adult nap duration and risk of all-cause mortality and cardiovascular diseases (CVD).

### Methods

PubMed, Cochrane Library, Embase and Web of Science databases were searched to identify eligible studies. The quality of observational studies was assessed using the Newcastle-Ottawa Scale. We performed all statistical analyses using Stata software version 14.0. For the meta-analysis, we calculated hazard ratio (HR) and their corresponding 95% confidence intervals (CIs). To assess publication bias, we used a funnel plot and Egger's test.

### Results

A total of 21 studies involving 371,306 participants revealed varying methodological quality, from moderate to high. Those who indulged in daytime naps faced a significantly higher mortality risk than non-nappers (HR: 1.28; 95% CI: 1.18–1.38; $I^2$ = 38.8%; P<0.001). Napping for less than 1 hour showed no significant association with mortality (HR: 1.00; 95% CI: 0.90–1.11; $I^2$ = 62.6%; P = 0.971). However, napping for 1 hour or more correlated with a 1.22-fold increased risk of mortality (HR: 1.22; 95% CI: 1.12–1.33; $I^2$ = 40.0%; P<0.001). The risk of CVD associated with napping was 1.18 times higher than that of non-nappers (HR: 1.18; 95% CI: 1.02–1.38; $I^2$ = 87.9%; P = 0.031). Napping for less than 1 hour did not significantly impact CVD risk (HR: 1.03; 95% CI: 0.87–1.12; $I^2$ = 86.4%; P = 0.721). However, napping for 1 hour or more was linked to a 1.37-fold increased risk of CVD (HR: 1.37; 95% CI: 1.09–1.71; $I^2$ = 68.3%; P = 0.007).

**Funding:** The author(s) received no specific funding for this work.

**Competing interests:** The authors have declared that no competing interests exist.

## Conclusions

Our meta-analysis indicates that taking a nap increases the risk of overall mortality and CVD mortality. It highlights that the long duration time of the nap can serve as a risk factor for evaluating both overall mortality and cardiovascular mortality.

## Introduction

Napping, also known as a daytime nap or short rest, involves taking a brief period of relaxation during the day, lasting from a few minutes to several hours. It is a widely practiced phenomenon globally, and an appropriately timed nap can mitigate health risks associated with insufficient sleep [1, 2]. Moreover, napping has been shown to enhance attention, improve work efficiency, and reduce stress [3–5]. Research indicates that napping can be associated with an increased risk of gastric cancer, hepatocellular carcinoma, and gastric adenocarcinoma [6–8], furthermore, a meta-analysis conducted by Cai (2023) revealed a close correlation between napping and the risk of obesity [9]. It also serves as a protective factor against cardiovascular disease (CVD), all-cause mortality, depression, and diabetes. However, the duration of napping appears pivotal in determining its effects. Napping influences both the autonomic nervous system (ANS) and the central nervous system, with prolonged naps potentially disrupting circadian rhythms [5, 10]. The interplay between these factors-nervous system effects, circadian disruption, all-cause mortality, and CVD—suggests a complex relationship where napping could be associated with both increased all-cause mortality and CVD risk [11, 12].

Exploring the relationship between napping, CVD, and all-cause mortality is crucial for predicting risk factors effectively and improving overall health outcomes. However, previous systematic reviews and meta-analyses have predominantly focused on nighttime sleep duration in relation to overall mortality, CVD, and associated risks [13, 14]. The connection between daytime napping and overall mortality, as well as cardiovascular risk factors, remains inconclusive. A meta-analysis conducted in 2015, comprising seven studies, suggested that napping could predict overall mortality but not cardiovascular mortality [15]. Contrarily, recent meta-analytical findings from 2020 diverge from earlier studies and omit seven recent publications [16–23]. Given the conflicting conclusions from these newly published studies and existing meta-analyses, the current evidence base may not be the most up-to-date. Therefore, this study aims to provide a comprehensive review of the latest evidence concerning the association between adult nap duration and both overall mortality and CVD.

## Methods

This meta-analysis was conducted according to the Preferred Reporting Items for Systematic Reviews and Meta-Analyses (PRISMA) statement [24] (S1 Table). The protocol has been registered in an International Prospective Register of Systematic Reviews with the registration number is CRD42024547547.

### Data sources and searches

We conducted a search for cohort studies published from the inception of the databases to April 30, 2024, in PubMed, Cochrane Library, Embase, and Web of Science. Our search was not limited by language, and we utilized a combination of medical subject headings (MeSH) and keywords, including terms like napping, siesta, dozing, catnap, snooze, mortality, death,

CVD, and heart failure. Detailed search strategies for each database are provided in S2 Table. Additionally, we manually screened the reference lists and relevant articles of all eligible studies to ensure comprehensive coverage.

### Eligibility criteria

We selected eligible studies meeting the following criteria: participants aged 18 years or older; population-based cohort studies with representative samples; assessment of daytime sleep time; investigation of all-cause or cause-specific mortality, CVD.

Exclusion criteria included: reports, reviews, conference abstracts, and studies reporting duplicate results; lack of odds ratio (OR) and hazard ratio (HR) estimates with corresponding 95% confidence intervals (CI).

### Study selection

Two researchers (MW and ZYZ) independently screened the literature based on the inclusion and exclusion criteria. After removing duplicates, the initial screening phase involved evaluating titles and abstracts. In the second phase, full texts of potentially eligible articles were reviewed to confirm adherence to the criteria. Any disagreements among the researchers were resolved through discussion with a third researcher (QHJ).

### Data extraction

Two researchers, WM and YC, followed established guidelines for data extraction and used a pre-designed table for this purpose. The extracted data included details such as the first author, publication year, country, follow-up duration, sample size, cohort characteristics, age distribution, assessment of daytime napping, subgroup analysis, adjustment for confounding factors, and NOS scores. Any discrepancies were resolved by consulting a third researcher (QHJ) (S3 Table).

### Risk of bias assessment

Based on the Newcastle-Ottawa Scale (NOS), two researchers assessed the quality of cohort studies, considering three key factors: participant selection for exposed and unexposed groups, comparability between the groups, and outcome assessment [25]. The total NOS score ranges from 0 to 9 points. Out of the articles included in this study, 6 were classified as having moderate quality, while the remaining articles were deemed high quality.

### Statistical analysis

We conducted a meta-analysis using the DerSimoniane-Laird random-effects model [26], comparing the prevalence of CVD and the overall risk ratio (HR) of all-cause mortality between the napping and non-napping populations. The non-napping group was chosen as the reference category. In cases where multiple adjusted estimates were reported in a study, we selected the estimate with the most adjustments. Additionally, we performed subgroup analysis based on gender. Sensitivity analysis was conducted to assess the robustness of the overall results. Publication bias was evaluated using a funnel plot and Egger's test [27]. All statistical analyses were performed using Stata software version 14.0.

## Results

### Literature search

Out of the 2422 studies identified in our search, 476 were excluded as duplicates. Following the screening of titles and abstracts, 1904 more studies were excluded. After full-text reading, 21 additional studies were excluded (S4 Table). Ultimately, 21 studies meeting the predefined inclusion criteria were included in the analysis. The selection process is depicted in Fig 1.

### Characteristics of the eligible studies

This meta-analysis included a total of 21 cohort studies, covering publication years from 1996 to 2024, with a total of 374,306 participants. These studies were conducted in 11 countries, with China having the highest number of studies, totaling 7 [17, 20–23, 28, 29], followed by the United States with 4 studies [30–33]. All participants in these cohorts were at least 18 years old at the start of the follow-up, and the follow-up period ranged from 4 to 18 years. Daytime napping assessments are commonly conducted using questionnaires and interviews. Although adjusted confounding factors may vary slightly, almost all studies provided adjusted estimates. Refer to Table 1 for the main characteristics of the included trials.

### Quality assessment

According to the NOS criteria, the average score for all included cohort studies was 7.52±0.87, with four studies scoring 6 [29, 30, 34, 35] and the remaining scores being 7 or higher. This indicates that the methodological quality of the included studies is moderate to high. The scores of the included studies are shown in Table 1 and S5 Table.

### Risk of all-cause mortality

A total of 10 eligible cohort studies [19, 21–23, 30–32, 34, 36, 37] investigated the relationship between napping and overall mortality risk. The results of the random-effects model analysis revealed a significantly higher risk of mortality in individuals who took daytime naps compared to those who did not (HR 1.28, 95%CI: 1.18–1.38; $I^2$ = 38.8%; P<0.001; Fig 2A). The risk of mortality associated with napping was 1.27 times higher than that of non-nappers. Further analysis was conducted to examine the relationship between nap duration and overall mortality risk by categorizing nap duration into <1 hour and ≥1 hour. Six studies were included in the analysis, and the forest plot indicated no significant association between napping for less than 1 hour and mortality risk (HR 1.00, 95% CI: 0.90–1.11; $I^2$ = 62.6%; P = 0.971; Fig 2B). However, the forest plot of seven studies showed that individuals who napped for 1 hour or more had a 1.22-fold increase in the risk of mortality compared to non-nappers (HR 1.22, 95%CI: 1.12–1.33; $I^2$ = 40.0%; P<0.001; Fig 2C). Sensitivity analysis indicated that no single study reversed the magnitude of the pooled effect, suggesting robustness of the results (S6 Table).

### Risk of cardiovascular mortality

A total of 12 cohort studies [17–20, 29, 31–33, 36, 38–40] met the inclusion criteria and examined the relationship between napping and CVD risk. The results of the random-effects model analysis showed that the CVD risk associated with napping was 1.18 times higher than that of non-nappers (HR 1.18, 95% CI: 1.02–1.38; $I^2$ = 87.9%; P = 0.031; Fig 2D). Further analysis was conducted to explore the relationship between nap duration and CVD risk. Six articles were included in the analysis, and there was no significant association between napping for less than 1 hour and CVD risk compared to not napping (HR 1.03, 95% CI: 0.87–1.12; $I^2$ = 86.4%;

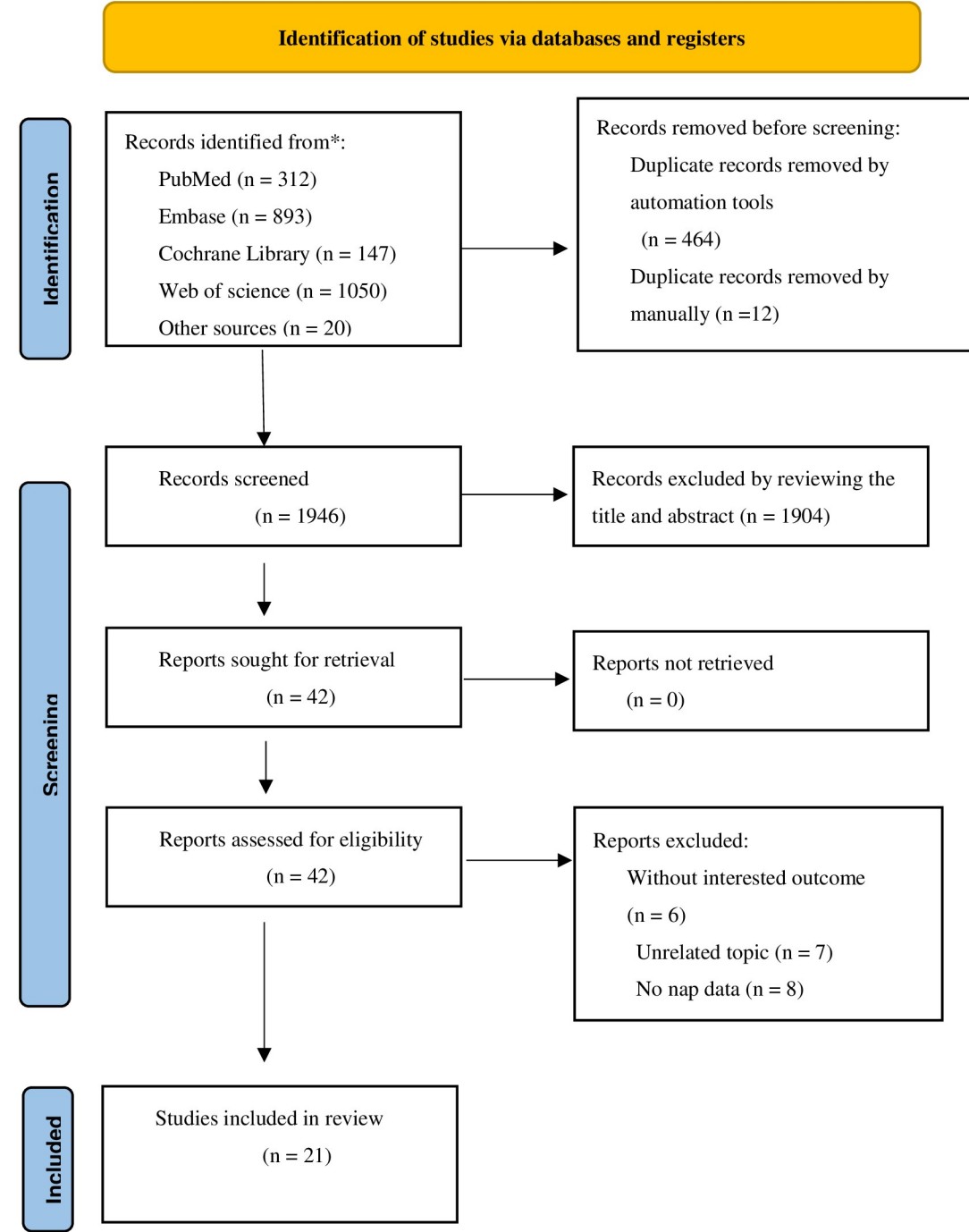

**Fig 1. Flow diagram of study selection and exclusion.**

P = 0.721; Fig 2E). However, individuals who napped for 1 hour or more had a 1.37-fold increase in cardiovascular risk compared to non-nappers (HR 1.37, 95%CI: 1.09–1.71; $I^2$ = 68.3%;P = 0.007; Fig 2F). Sensitivity analysis indicated that none of the individual studies reversed the magnitude of the combined effect, suggesting robustness of the results (S6 Table).

**Table 1. Characteristics of the cohort studies included in the meta-analysis.**

| Autor | Year | Country | Follow-up (years) | No. of participants | Age | Cohort | Assessment of daytime napping | Subgroup analysis | NOS scores | Confounders adjusted |
|---|---|---|---|---|---|---|---|---|---|---|
| Hays, J, C. | 1996 | US | 4 | 3,962 | 65–101 | EPESE | Questionnaire | / | 6 | / |
| Bursztyn, M | 1999 | Israel | 6.5 | 455 | 70 | Ministry of Interior Registry | Interview | Siesta, no siesta | 6 | / |
| Bursztyn, M. | 2002 | Jerusalem | 6 | 442 | 70 | Ministry of Interior Registry | Questionnaire | All men, women | 6 | / |
| Naska, A. | 2007 | Greek | 6.32 | 28,571 | 20–86 | European Prospective Investigation into Cancer and Nutrition (EPIC) | Questionnaires | Men, women | 7 | / |
| Stone, K, L. | 2009 | US | 6.9 | 8,101 | 77 | SOF | Questionnaire | / | 8 | age; body mass index; history of at least one medical condition including diabetes mellitus, Parkinson's disease, dementia, chronic obstructive pulmonary disease, non-skin cancer, and osteoarthritis; history of CVD; history of hypertension; walks for exercise; alcohol use; smoking status; depression; cognitive impairment; estrogen use; and benzodiazepine use |
| Tanabe, N. | 2010 | Japan | 14.3 | 67,129 | 40–79 | JACC study | Self-administered questionnaire | All men, women | 8 | Sex, age, sleeping duration, treated hypertension, history of diabetes, any disease under medical treatment, smoking status, BMI, weight loss from age 20 years, blood pressure, perceived mental stress, depressive symptoms, working status, educational status and time for walking |

*(Continued)*

**Table 1.** (Continued)

| Autor | Year | Country | Follow-up (years) | No. of participants | Age | Cohort | Assessment of daytime napping | Subgroup analysis | NOS scores | Confounders adjusted |
|---|---|---|---|---|---|---|---|---|---|---|
| Stang, A. | 2012 | Germany | 8.1 | 4,123 | 45–75 | the Heinz Nixdorf Recall study | Interview | Men, women, men and women | 8 | including smoking (never, former, current) and pack-years smoked, history of diabetes mellitus, regular difficulties falling asleep (dichotomous variables) and age, systolic and diastolic blood pressure at baseline, BMI, waist circumference, log(CAC+1), CRP, LDL and HDL cholesterol, ankle-brachial index, pack-years smoked, and sleep duration at night |
| Leng,Y. | 2014 | UK | 13 | 16,374 | 40–79 | EPIC-Norfolk prospective cohort study | Questionnaire | Length of follow-up, Age, Sex, Employment status, preexisting health conditions, Smoking status, Category of body mass index, Major depressive disorder, Time spent in bed at night, Nighttime sleep duration | 8 | ge and sex. social class, educational level, marital status, employment status, body mass index, physical activity level, smoking status, and alcohol intake. depression, self-reported general health, hypnotic drug use, antidepressant use, chronic obstructive pulmonary disease drug use, and time spent in bed at night. self-reported preexisting diseases and underlying sleep apnea |
| Wannamethee, S,G. | 2016 | UK | 9 | 3,723 | 40–59 | The British Regional Heart study | Questionnaire | All, No preexisting CVD, Preexisting CVD | 8 | age, type of work, body mass index, smoking, diabetes mellitus, physical activity, treated hypertension, breathlessness, preexisting myocardial infarction, stroke, poor health. age, social class, body mass index, smoking, physical activity, diabetes mellitus, treated hypertension, breathlessness, and poor health |

*(Continued)*

**Table 1.** (Continued)

| Autor | Year | Country | Follow-up (years) | No. of participants | Age | Cohort | Assessment of daytime napping | Subgroup analysis | NOS scores | Confounders adjusted |
|---|---|---|---|---|---|---|---|---|---|---|
| Zhou,J,M. | 2016 | China | / | 13,469 | ≥40 | Chinese Family Panel Studies (CFPS) | Interview | Women, Men | 7 | age, sex, race, marital status, education level, employment, annual household income, self-rated social status, community types, smoking, alcohol consumption, self-rated health status, health insurance, remember important things within one week, physical activity, body mass index (BMI), and depression |
| Wang,C.S. | 2017 | China | / | 46,285 | 35–70 | Prospective Urban Rural Epidemiology (PURE) | Interviews, anthropometric and biochemical measurements | CVD, Stroke, Coronary artery disease | 6 | age, gender, body mass index, education attainment, marital status, smoking, drinking, physical activity, region, total cholesterol, diabetes, hypertension, and depression |
| Xiao,Q. | 2017 | US | 7.8 | 4,869 | 64.6 | The NIH-AARP Diet and Health Study | Questionnaire | All-cause death, CVD death | 9 | age, sex, cancer site, tumor stage, tumor grade, surgery, chemotherapy, radiation, education, smoking, TV viewing, MVPA, BMI, self-reported health, history of heart disease, stroke and diabetes, napping |
| Häusler,N. | 2019 | Swiss | 5.3 | 3,462 | 35–75 | CoLaus | Interview, Questionnaire | Nap frequency, Average daily nap duration over a week | 8 | age, sex, education, smoking status, sedentary behaviour (yes/no), BMI (normal, overweight, obese) and sleep duration. hypertension, diabetes and dyslipidaemia |
| Yan,B. | 2019 | US | 11 | 4,170 | 63.1 | SHHS database | Medical records, interview, Questionnaire | CVD. Hypertension, Non-hypertension, | 8 | age, sex, daytime napping, race, education, marital status, smoking status, diabetes mellitus, hypertension, sleep duration, neck circumference, waist circumference, triglyceride |
| Wang,L. | 2022 | China | 5 | 42,590 | Adults | CFPS | Interview, | CVDs, HTN, stroke | 8 | gender, Age, BMI, Residentialregion, Geolocation, Physical activity, Smoking status, Alcohol consumption, Sleep duration |

*(Continued)*

**Table 1.** (Continued)

| Autor | Year | Country | Follow-up (years) | No. of participants | Age | Cohort | Assessment of daytime napping | Subgroup analysis | NOS scores | Confounders adjusted |
|---|---|---|---|---|---|---|---|---|---|---|
| Wang,Z,Y. | 2022 | Sweden | 18 | 12,268 | 70.3 | STR (Swedish Twin Registry), | Telephone interview. | / | 8 | sex and education, marital status, body mass index, smoking status, alcohol consumption, physical activity, type 2 diabetes, hypertension, and depression |
| Chen,A. | 2023 | British | 9 | 1,722 | 78.55 | British Regional Heart Study (BRHS) | Self-report | All-cause mortality, Cardiovascular mortality, Non-cardiovascular mortality | 8 | Age, smoking, physical activity, and BMI. diabetes, frailty, general health, antihypertensive medication |
| Diao,T,Y. | 2023 | China | 7.2 | 31,500 | 61.2 | Dongfeng-Tongji (DFTJ) cohort. | Questionnaire | CVD, CHD, Stroke | 8 | age, sex, year of recruitment, education level, smoking status, drinking status, regular exercise, body mass index, hypertension, diabetes, hyperlipidemia, and family history of CVD, CHD, or stroke |
| Ke,W. | 2023 | China | 7.1 | 15,524 | ≥45 | Chinese Health and Retirement Longitudinal Study (CHARLS). | Interview | / | 8 | sex, age, BMI, marital status, residential region, education attainment, social activity, smoking status, alcohol consumption and chronic disease |
| Wang,L. | 2023 | China | 8 | 41,950 | adults | China Family Panel Studies (CFPS) | Interviews | all-cause mortality | 7 | gender, age, BMI, ethnicity, marital status, education attainment, employment status, household income, urbanization, geographical region, physical activity, smoking status, alcohol consumption, chronic diseases, depressive symptoms and sleep duration |
| Zhang,Y,T. | 2024 | China | 7 | 20,617 | ≥45 | CHARLS | Interview | All-cause mortality, Premature mortality | 8 | age, gender, body mass index, area, marital status, education, smoking, alcohol intake, 10-item Center for Epidemiological Studies Depression Scale scores, hypertension, diabetes, cancer, CVD, and night sleep duration/daytime napping duration if application |

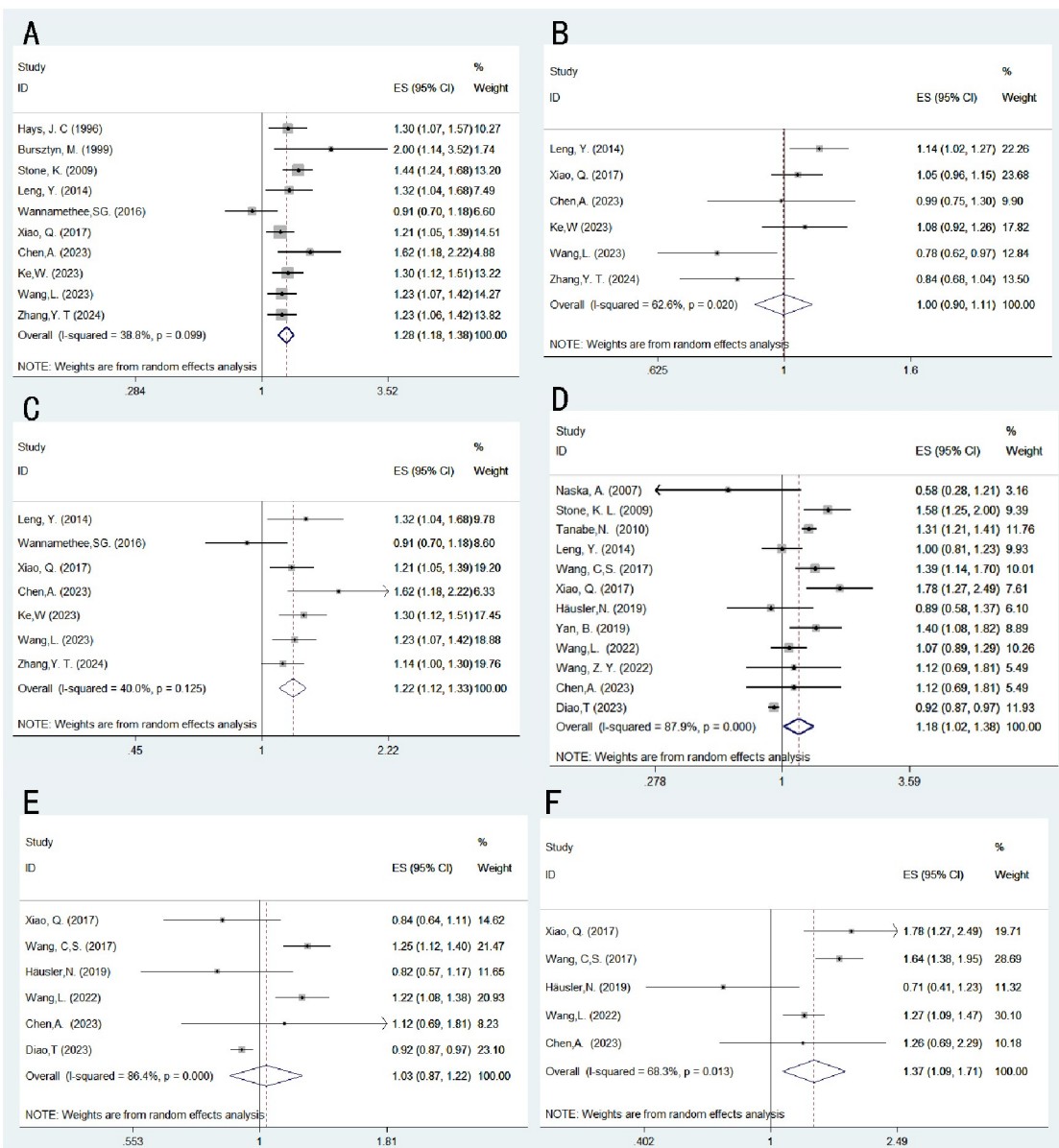

**Fig 2. Illustrates the forest plots for the association between daytime napping and all-cause mortality, as well as CVD risk.** (A) napping and non-napping in relation to the risk of all-cause mortality. (B) napping less than 1 hour and non-napping in relation to the risk of all-cause mortality. (C) napping 1 hour or longer and non-napping in relation to the risk of all-cause mortality. (D) daytime napping and non-napping in relation to CVD risk. (E) napping less than 1 hour and non-napping in relation to CVD risk. (F) napping 1 hour or longer and non-napping in relation to CVD risk.

## Subgroup analysis

We conducted a subgroup analysis based on gender, and the results showed that there was no effect of daytime napping on all-cause mortality for men (HR, 1.04; 95% CI: 0.81–1.33; $I^2$ = 0%; P = 0.795). Similarly, there was no effect of daytime napping on all-cause mortality for women (HR, 1.07; 95% CI: 0.83–1.38; $I^2$ = 0%; P = 0.601). On the other hand, we found that neither men (HR, 1.10; 95% CI: 0.87–1.40; $I^2$ = 10.9%;P = 0.436) nor women (HR, 1.08; 95%

**Table 2. Subgroup analyses of all-cause mortality with and without napping.**

| Subgroups | Included studies | OR (95%CI) | Heterogeneity | |
|---|---|---|---|---|
| | | | I$^2$(%) | P-values |
| **All-cause mortality** | | | | |
| Male | 4 | 1.04(0.81–1.33) | 0 | 0.847 |
| Female | 4 | 1.07(0.83–1.38) | 0 | 0.907 |
| **Cardiovascular mortality** | | | | |
| Male | 4 | 1.10(0.87–1.40) | 10.9 | 0.338 |
| Female | 3 | 1.08(0.76–1.54) | 0 | 0.605 |

CI: 0.81–1.33; I$^2$ = 0%; P = 0.652) showed an effect of daytime napping on cardiovascular mortality (Table 2).

## Publication bias

A visual inspection of the funnel plot showed no evidence of a significant publication bias in the outcome of risk of all-cause mortality with and without napping, risk of cardiovascular mortality associated with napping compared with not napping (Fig 3). Egger's regression test (P>0.05) likewise indicated no publication bias in our meta-analysis.

## Discussion

### Main findings

This meta-analysis comprises 21 cohort studies involving 371,306 individuals, offering a comprehensive assessment of the relationship between napping and both overall mortality and CVD. We identified statistically significant associations indicating increased risks of overall mortality and CVD among adults who nap compared to those who do not. Specifically, napping was associated with a 1.28-fold higher risk of overall mortality and a 1.18-fold higher risk of CVD. Furthermore, napping for less than 1 hour did not show a significant association with either outcome. In contrast, napping for 1 hour or more was linked to a 1.22-fold higher risk of overall mortality and a 1.37-fold higher risk of CVD compared to non-nappers. Interestingly, gender did not influence the observed associations between daytime napping and the risks of overall mortality or cardiovascular disease.

### Interpretation of findings

A 2015 meta-analysis synthesized findings from 7 cohort studies that explored the connection between daytime nap duration and overall mortality as well as CVD [15]. It revealed that individuals who nap for more than 60 minutes per day appear to face a higher risk of overall mortality compared to those who do not nap. However, while napping may predict overall mortality, it does not seem to predict cardiovascular mortality. These findings contrast with our current study. One possible reason for this discrepancy could be the smaller number of studies included in the 2015 meta-analysis. Moreover, the majority of studies in our research were conducted more recently, reflecting potential changes in era and lifestyle habits. Another similar meta-analysis, incorporating 13 studies, corroborated our findings by highlighting a significant association between prolonged nap duration (over 60 minutes) and heightened risks of overall mortality and CVD [16]. To obtain more comprehensive and up-to-date insights into the relationship between nap duration and these health outcomes, our study

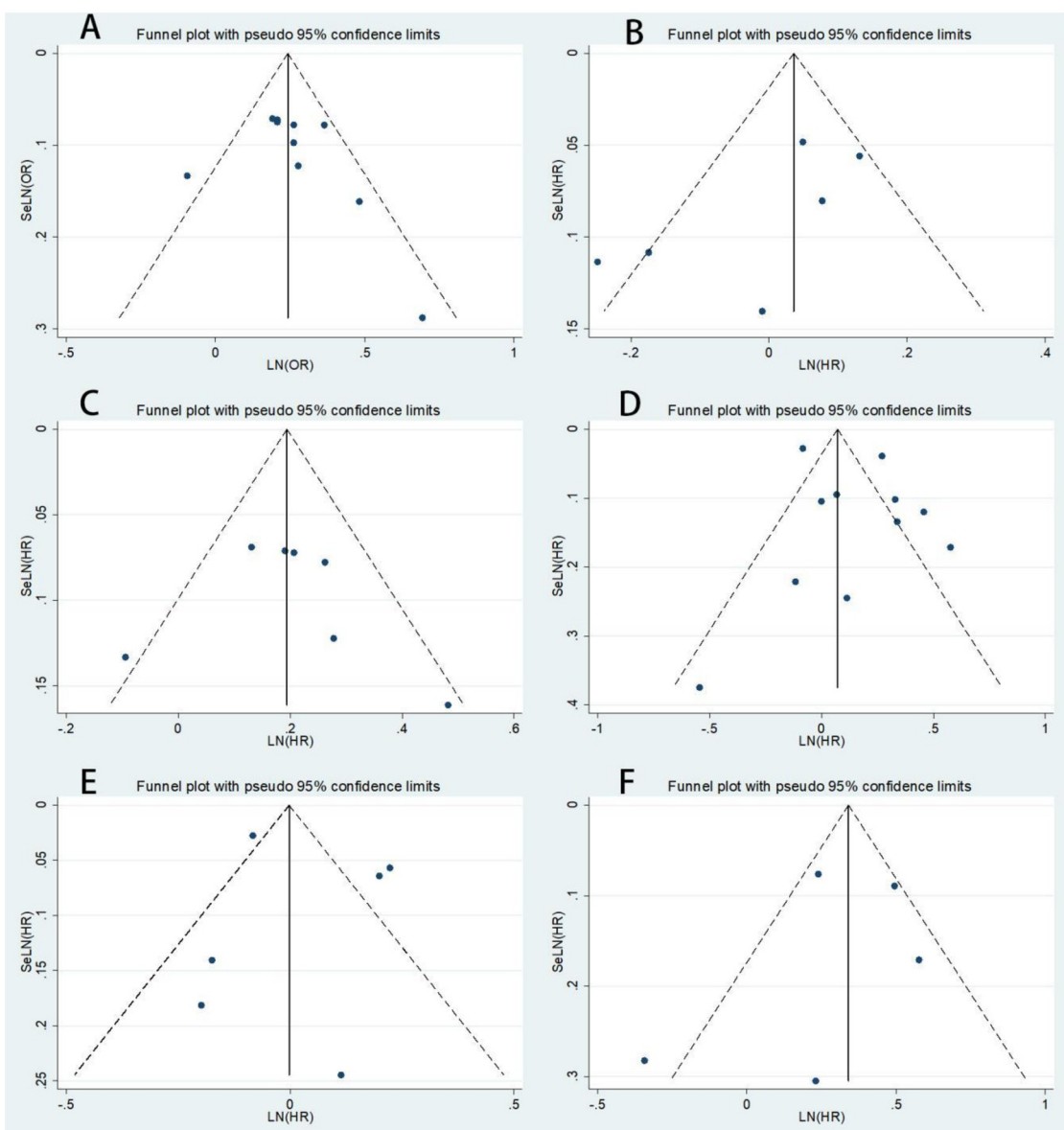

**Fig 3. Funnel plot showing the effect of different nap durations on all-cause mortality and CVD.**

systematically reviewed relevant research, integrating 7 additional recent studies from the past few years.

The findings of this study indicate that gender does not appear to be a significant variable in assessing the impact of daytime napping on overall mortality and CVD risk, aligning with the conclusions of another meta-analysis [15]. However, a prospective cohort study found a significant independent correlation between napping for more than 30 minutes during the day and overall mortality in men, whereas this association was not observed in women [41]. This disparity may be attributed to the cohort study's focus on older adults, potentially influencing sleep patterns and physiological conditions differently between older adults and younger adults. Another study highlighted a notable link between longer nap durations among older

women and an increased susceptibility to cardiovascular disease [42]. Additionally, Wang et al.'s research revealed a marked rise in overall mortality risk associated with naps exceeding 60 minutes in adult males [22]. These varied research outcomes suggest a possible gender-specific relationship between daytime napping and the risks of overall mortality and CVD, necessitating further investigation and exploration.

Excessive daytime napping of an hour or more is associated with an increased risk of CVD. This could be due to changes in body position post-nap, which might trigger cardiovascular incidents [43], or an elevation in heart rate linked to heightened sympathetic nerve activity upon waking [44]. Moreover, research has established a causal link between daytime napping and cardiac metabolic diseases, underscoring daytime sleepiness as a potential risk factor for such conditions [45]. Additionally, prolonged daytime napping correlates with higher rates of diabetes, Alzheimer's disease, cancer, and obesity [9, 46–48]. The exact pathological mechanisms behind these adverse effects remain incompletely understood but likely involve napping's impact on various physiological systems. For example, excessive napping can affect brain volume, hippocampal size, and potentially lead to the presence of Lewy bodies and loss of substantia nigra neurons [49, 50]. It may also reduce insulin sensitivity [51], all contributing to increased mortality risk. It's noteworthy that some evidence suggests longer daytime naps could benefit cardiovascular health [52], though this finding stems from research involving young adults aged 18 to 35 years. Further investigation is necessary to determine if age influences these outcomes differently. There is a certain association between excessive daytime napping and health conditions. Some studies have indicated that physically weak individuals have significantly reduced daytime activity and increased time spent in bed, leading to a higher likelihood of experiencing sleepiness [53]. Additionally, research has found that individuals with poorer health conditions engage in less long-term napping [54]. On the other hand, sleep quality also affects the duration of daytime napping. For example, sleep apnea may cause excessive daytime sleepiness [55]. Studies have shown that individuals who nap are more likely to experience sleep apnea, which is one of the reasons for cardiovascular diseases in this population, special attention should be given to individuals who experience breathing pauses [56, 57]. Further research can delve into the relationship between different sleep disorders and daytime napping in order to enhance our understanding of the association between daytime napping and health conditions.

The findings from this study indicate that daytime naps lasting less than 1 hour are not associated with an increased overall mortality rate or risk of CVD, consistent with a previous meta-analysis [16]. This view is further supported by the study conducted by Wang et al. [21]. Importantly, there are findings suggesting that short naps may actually reduce mortality and lower cardiovascular disease risk [23, 58]. Conversely, other research suggests that napping for less than 30 minutes may have a protective effect on overall mortality and CVD risk, whereas napping for 30–60 minutes could potentially increase these risks [17, 22]. Moving forward, future studies could aggregate data to provide a more nuanced classification of naps shorter than 1 hour, delving deeper into their specific impacts on overall mortality and cardiovascular disease.

## Implications and limitations

Our meta-analysis consolidates current evidence on the association between napping and overall mortality rate as well as CVD risk, underscoring the significance of adult nap duration in early detection of these health outcomes. However, this study is subject to certain limitations. Due to limited data, we did not conduct subgroup analysis based on different regions and races. We exclusively analysed cohort studies; future investigations could benefit from integrating case-control and cross-sectional studies to diversify study methodologies.

Moreover, variations in lifestyle factors and sleep habits across different countries might influence research findings. Additionally, our meta-analysis did not incorporate covariate analysis, although the cohort studies included in our review controlled for confounding variables, ensuring the robustness of our conclusions. It is important to acknowledge that reliance on interviews or questionnaires to assess nap duration could introduce measurement errors and recall biases.

## Conclusions

This meta-analysis indicates that prolonged napping is associated with increased risks of overall mortality and cardiovascular disease. However, further research is necessary to confirm the underlying pathophysiological mechanisms involved, which could include longitudinal observational studies and genetic investigations. Additionally, to assess whether nap duration can reliably predict the risks of overall mortality and cardiovascular disease, more prospective studies involving adults are needed.

## Supporting information

**S1 Table. PRISMA checklist.**
(DOCX)

**S2 Table. Retrieval strategies.**
(DOCX)

**S3 Table. Data extraction information.**
(XLSX)

**S4 Table. Literature exclusion form.**
(XLSX)

**S5 Table. Quality assessment.**
(DOCX)

**S6 Table. Sensitivity analysis.**
(DOCX)

## Acknowledgments

The authors would like to express their gratitude to all the experts for their guidance, professional knowledge, and valuable advice throughout the entire process. They have provided selfless guidance and assistance during the entire research process. Their expertise and experience have played a significant role in driving our research forward.

## Author Contributions

**Conceptualization:** Meng Wang, Yang Cao.

**Data curation:** Xin Xiang, Zhengyan Zhao, Linlin Hou.

**Investigation:** Meng Wang, Yang Cao.

**Methodology:** Meng Wang, Yu liu, Yang Cao.

**Resources:** Qiuhuan Jiang.

**Validation:** Zhengyan Zhao, Weiwei Guo, Qiuhuan Jiang.

**Writing – original draft:** Meng Wang.

**Writing – review & editing:** Meng Wang, Xin Xiang, Yu liu, Weiwei Guo, Linlin Hou, Qiu-huan Jiang.

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
