## [Decision Letter · Decision Letter 0]

27 Aug 2024

PONE-D-24-26341Association between self-reported napping and risk of cardiovascular disease and all-cause mortality: A meta-analysis of cohort studiesPLOS ONE

Dear Dr. Jiang,

Thank you for submitting your manuscript to PLOS ONE. After careful consideration, we feel that it has merit but does not fully meet PLOS ONE’s publication criteria as it currently stands. Therefore, we invite you to submit a revised version of the manuscript that addresses the points raised during the review process.

We look forward to receiving your revised manuscript.

Kind regards,

Nagarajan Raju

Academic Editor

PLOS ONE

Journal requirements: 1. When submitting your revision, we need you to address these additional requirements. Please ensure that your manuscript meets PLOS ONE's style requirements, including those for file naming. The PLOS ONE style templates can be found at https://journals.plos.org/plosone/s/file?id=wjVg/PLOSOne_formatting_sample_main_body.pdf and https://journals.plos.org/plosone/s/file?id=ba62/PLOSOne_formatting_sample_title_authors_affiliations.pdf. 2. Please include captions for your Supporting Information files at the end of your manuscript, and update any in-text citations to match accordingly. Please see our Supporting Information guidelines for more information: http://journals.plos.org/plosone/s/supporting-information. 

Additional Editor Comments:

I suggest authors to go through all the comments from the reviewers and address those in the revised version of the manuscript

Reviewers' comments:

Reviewer's Responses to Questions

**Comments to the Author**

1. Is the manuscript technically sound, and do the data support the conclusions?

Reviewer #1: Yes

Reviewer #2: Yes

2. Has the statistical analysis been performed appropriately and rigorously? 

Reviewer #1: Yes

Reviewer #2: Yes

3. Have the authors made all data underlying the findings in their manuscript fully available?

Reviewer #1: Yes

Reviewer #2: Yes

4. Is the manuscript presented in an intelligible fashion and written in standard English?

Reviewer #1: Yes

Reviewer #2: Yes

5. Review Comments to the Author

Reviewer #1: This is very interesting study. Usually It is considered napping relax the body , refresh mind and increase the work efficiency but this study claims that day time napping increases the chances of heart problems. Few concerns;

Major points.

Does the authors considered the race of the people on which these study was done? I think major countries include China and US as per authors, Do you see any difference between people of Asian origin or American?

Gender based subgroup analysis is a little unclear. It needs further details.

What you think for the people which has to sleep over the day time for few hours such as folks who works over night shifts ?

Minor points;

The authors stated one statement as ‘ These studies were conducted in 11 countries, with China having the highest number of studies, totaling 7 (15, 18-21, 26, 27), followed by the United States with 4 studies (28-31) ‘

Is it 11 countries or 11 states. It is a little confusing statement.

Another statement ‘. Research indicates that while napping can be a risk factor for cancer and obesity ‘ needs more description in the introduction part.

Reviewer #2: The paper deals with a very complex subject because the articles used for the meta-analysis are very heterogeneous in terms of the age of the sample and, above all, the way in which the time spent napping is counted, based on frequency and duration. It would be expected that two three-hour naps a week would not have the same biological effect as a one-hour daily nap from Monday to Saturday.

The authors have tried to use the appropriate statistical methods for meta-analysis, although some authors, such as Stang (2010), have questioned the validity of the NOS Newcastle-Ottawa Scale procedure, arguing that it can lead to arbitrary results. In any case, as far as I know, Wells et al. consider studies that achieve a NOS score of seven or more to be valid. Please state more clearly what the cut-off point was for discarding a study.

Stang A. Critical evaluation of the Newcastle-Ottawa scale for the assessment of the quality of nonrandomized studies in meta-analyses. Eur J Epidemiol 2010;25(9):603–605.

Although some associations are significant, the HR values are quite low and the IQ variations are in some cases very small. It is clear that the observed association does not imply directionality, so one might ask what elements might be acting as confounders beyond those that could be controlled for.

Could it be that people in poorer health are more likely to nap? On the other hand, could there be a link between poorer sleep (e.g. due to apnoea problems) and a greater tendency to nap?

These considerations do not invalidate your work, but I think they should be added to the discussion or to the limitations section.

6. PLOS authors have the option to publish the peer review history of their article (what does this mean?). If published, this will include your full peer review and any attached files.

Reviewer #1: No

Reviewer #2: **Yes: **María Dolores Marrodán Serrano

---

## [Author Response · Author response to Decision Letter 0]

3 Sep 2024

Dear reviewers and editor:

On behalf of my co-authors, we thank you very much for your letter and comments on our manuscript entitled "Association between self-reported napping and risk of cardiovascular disease and all-cause mortality: A meta-analysis of cohort studiess" (Manuscript ID: PONE-D-24-26341). We appreciate the editor and reviewers for their constructive and valuable comments. We have revised our manuscript considerably according to the editors’ and reviewers’ comments, questions, and suggestions. In the event that we missed any one of the comments please let us know. This document includes our responses to reviewers and editor comments point by point, and the revised portion are marked in RED in our manuscript. 

Reply to reviewer 1

Comment 1: This is very interesting study. Usually It is considered napping relax the body, refresh mind and increase the work efficiency but this study claims that day time napping increases the chances of heart problems. 

Reply 1: Thank you very much for your affirmation and interest in this topic. Based on clinical observations and literature review, we have decided to conduct this meta-analysis. The results indicate an association between self-reported napping and overall mortality and cardiovascular diseases. Through this quantitative analysis, we can to some extent change certain clinical practice decisions, which is the significance of this meta-analysis. Thank you again for your affirmation. We are committed to publishing more excellent articles and look forward to your further guidance.

Comment 2:

Does the authors considered the race of the people on which these study was done? I think major countries include China and US as per authors, do you see any difference between people of Asian origin or American?

Reply 2: Thank you for your attention. We have conducted a reanalysis based on the original literature. Among them, 7 studies are from China and 4 studies are from the United States. Unfortunately, these studies cannot be subjected to subgroup analysis because they do not provide independent data on the relationship between different napping durations and the risk of overall mortality and cardiovascular events. Meanwhile, we look forward to the emergence of more in-depth analysis results based on different races. Once these results are available, we further update our analysis. Of course, the point you mentioned may also be a limitation of this study. In the limitation section, we have mentioned that due to limited data, we did not conduct subgroup analysis based on different regions and races. We look forward to future studies being able to further perform subgroup analysis based on different variables.

Change in revised text: 

Page 15, lines 310-311, in red.

Due to limited data, we did not conduct subgroup analysis based on different regions and races.

Comment 3: Gender based subgroup analysis is a little unclear. It needs further details.

Reply 3: We analyzed the differences in daytime napping on the outcomes of overall mortality and cardiovascular diseases in different genders. Due to the limitations of the scope, we were only able to analyze these two dimensions. We found no statistically significant differences in overall mortality risk and cardiovascular diseases risk between males and females. Looking forward to your understanding and support.

Comment 4: What you think for the people which has to sleep over the day time for few hours such as folks who works over night shifts?

Reply 4: This is a valid concern and we have taken it into consideration while writing this meta-analysis. However, most included studies exclude the population of night shift workers because our study does not consider the need for additional daytime sleep among night shift workers. This exclusion is to avoid the confounding effect of night shift work on the results and ensure the robustness of our findings.

Comment 5: The authors stated one statement as ‘These studies were conducted in 11 countries, with China having the highest number of studies, totaling 7 (15, 18-21, 26, 27), followed by the United States with 4 studies (28-31) ‘Is it 11 countries or 11 states. It is a little confusing statement.

Reply 5: Thank you for pointing out the confusion. These studies were indeed conducted in 11 countries, not states.

Change in revised text: Page 5, lines 120-122, in red.

These studies were conducted in 11 countries, with China having the highest number of studies, totaling 7 (17, 20-23, 28, 29), followed by the United States with 4 studies (30-33).

Comment 6: Another statement ‘Research indicates that while napping can be a risk factor for cancer and obesity ‘needs more description in the introduction part.

Reply 6: Thank you for your constructive suggestion. We have expanded on the relevant research findings in the introduction section, which will provide readers with a clearer and more comprehensive understanding of the topic. Thank you for bringing this to our attention.

Change in revised text: Page 2, lines 44-46, in red.

Research indicates that napping can be associated with an increased risk of gastric cancer, hepatocellular carcinoma, and gastric adenocarcinoma(6-8), furthermore, a meta-analysis conducted by Cai (2023) revealed a close correlation between napping and the risk of obesity (9). 

We really appreciate your positive and insightful suggestions on our manuscript. Under your kind help and professional guidance, we believe that our manuscript has been improved substantially. We are looking forward to your further suggestions.

Reply to reviewer 2

Comment 1: The paper deals with a very complex subject because the articles used for the meta-analysis are very heterogeneous in terms of the age of the sample and, above all, the way in which the time spent napping is counted, based on frequency and duration. It would be expected that two three-hour naps a week would not have the same biological effect as a one-hour daily nap from Monday to Saturday.

Reply 1: Thank you very much for your careful review. As you mentioned, we also noticed these issues when conducting this meta-analysis, such as inconsistencies in patient age, nap frequency, duration, and interval. Considering this situation, we can only choose samples with stronger consistency for quantitative meta-analysis, which makes the results more robust and reliable. We have strived to present the research results in a more appropriate quantitative form in order to achieve rigor and scientific validity. We look forward to your further feedback and guidance.

Comment 2: The authors have tried to use the appropriate statistical methods for meta-analysis, although some authors, such as Stang (2010), have questioned the validity of the NOS Newcastle-Ottawa Scale procedure, arguing that it can lead to arbitrary results. In any case, as far as I know, Wells et al. consider studies that achieve a NOS score of seven or more to be valid. Please state more clearly what the cut-off point was for discarding a study.

Stang A. Critical evaluation of the Newcastle-Ottawa scale for the assessment of the quality of nonrandomized studies in meta-analyses. Eur J Epidemiol 2010;25(9):603–605.

Reply 2: Thank you for your concern. This is a very interesting topic that deserves further discussion. Our meta-analysis included 21 articles, with only a few articles having an NOS score of 6, while the majority of articles had a score of 7 or higher, indicating their high quality. Additionally, we did not exclude studies of moderate quality, and there is a reason for this. According to the requirements in the Cochrane Handbook, during the stage of literature selection, articles cannot be excluded based solely on their quality, unless sensitivity analysis reveals significant heterogeneity caused by these low-quality studies. Only under such circumstances would we exclude these studies in order to better avoid selection bias. We believe that you are also an expert in meta-analysis, and our common goal is to minimize bias as much as possible, making the results more reliable and robust. We look forward to your further suggestions.

Comment 3: Although some associations are significant, the HR values are quite low and the IQ variations are in some cases very small. It is clear that the observed association does not imply directionality, so one might ask what elements might be acting as confounders beyond those that could be controlled for.

Reply 3: Thank you for raising such a precise and valuable topic. Although we have found a certain association between daytime napping and risk of overall mortality and cardiovascular disease, the effect size of this association is small. Additionally, the precision of the results is similar to the range of IQ, indicating a high level of reliability. We accepted the current findings while recognizing the need to control for many confounding biases. In addition to the factors you mentioned, we also consider that these studies encompass different races and countries, which may be another source of bias. Therefore, in the limitations section of the article, we stated that due to limited data, we did not conduct subgroup analysis based on different regions and races. We look forward to further research that can delve deeper into subgroup analysis based on different variables.

Change in revised text:

Page 15, lines 310-311, in red.

Due to limited data, we did not conduct subgroup analysis based on different regions and races.

Comment 4: Could it be that people in poorer health are more likely to nap? On the other hand, could there be a link between poorer sleep (e.g. due to apnoea problems) and a greater tendency to nap?

These considerations do not invalidate your work, but I think they should be added to the discussion or to the limitations section.

Reply 4: Thank you for raising a very valuable question. We have consulted detailed literature and discussed the relationship between poor health status, sleep quality, and napping in the discussion section.

Change in revised text:

Page 14, lines 285-297, in red.

There is a certain association between excessive daytime napping and health conditions. Some studies have indicated that physically weak individuals have significantly reduced daytime activity and increased time spent in bed, leading to a higher likelihood of experiencing sleepiness(53). Additionally, research has found that individuals with poorer health conditions engage in less long-term napping(54). On the other hand, sleep quality also affects the duration of daytime napping. For example, sleep apnea may cause excessive daytime sleepiness(55). Studies have shown that individuals who nap are more likely to experience sleep apnea, which is one of the reasons for cardiovascular diseases in this population, special attention should be given to individuals who experience breathing pauses(56, 57). Further research can delve into the relationship between different sleep disorders and daytime napping in order to enhance our understanding of the association between daytime napping and health conditions.

We really appreciate your positive and insightful suggestions on our manuscript. Under your kind help and professional guidance, we believe that our manuscript has been improved substantially. We are looking forward to your further suggestions.

Sincerely yours,

Qiuhuan Jiang

Email: qiuhuan1890@163.com

---

## [Decision Letter · Decision Letter 1]

17 Sep 2024

Association between self-reported napping and risk of cardiovascular disease and all-cause mortality: A meta-analysis of cohort studies

PONE-D-24-26341R1

Dear Dr. Jiang,

We’re pleased to inform you that your manuscript has been judged scientifically suitable for publication and will be formally accepted for publication once it meets all outstanding technical requirements.

Kind regards,

Nagarajan Raju

Academic Editor

PLOS ONE

Additional Editor Comments (optional):

Reviewers' comments:

Reviewer's Responses to Questions

**Comments to the Author**

1. If the authors have adequately addressed your comments raised in a previous round of review and you feel that this manuscript is now acceptable for publication, you may indicate that here to bypass the “Comments to the Author” section, enter your conflict of interest statement in the “Confidential to Editor” section, and submit your "Accept" recommendation.

Reviewer #1: All comments have been addressed

Reviewer #2: All comments have been addressed

2. Is the manuscript technically sound, and do the data support the conclusions?

Reviewer #1: Yes

Reviewer #2: Yes

3. Has the statistical analysis been performed appropriately and rigorously? 

Reviewer #1: Yes

Reviewer #2: Yes

4. Have the authors made all data underlying the findings in their manuscript fully available?

Reviewer #1: Yes

Reviewer #2: Yes

5. Is the manuscript presented in an intelligible fashion and written in standard English?

Reviewer #1: Yes

Reviewer #2: Yes

6. Review Comments to the Author

Reviewer #1: (No Response)

Reviewer #2: Dear authors, the changes made have improved the article and in my opinion it has reached the sufficient quality to be published. I congratulate you and wish you the best of luck in the future.

7. PLOS authors have the option to publish the peer review history of their article (what does this mean?). If published, this will include your full peer review and any attached files.

Reviewer #1: No

Reviewer #2: No

---

## [Editor Report · Acceptance letter]

7 Oct 2024

PONE-D-24-26341R1 

PLOS ONE

Dear Dr. Jiang, 

I'm pleased to inform you that your manuscript has been deemed suitable for publication in PLOS ONE. Congratulations! Your manuscript is now being handed over to our production team.

Kind regards, 

on behalf of

Dr. Nagarajan Raju 

Academic Editor

PLOS ONE